# Scaling Autoregressive Video Models

**Dirk Weissenborn**[*]
Google Research
diwe@google.com

**Oscar Täckström**[*†]
Sana Labs
oscar@sanalabs.com

**Jakob Uszkoreit**
Google Research
usz@google.com

## Abstract

Due to the statistical complexity of video, the high degree of inherent stochasticity, and the sheer amount of data, generating natural video remains a challenging task. State-of-the-art video generation models often attempt to address these issues by combining sometimes complex, usually video-specific neural network architectures, latent variable models, adversarial training and a range of other methods. Despite their often high complexity, these approaches still fall short of generating high quality video continuations outside of narrow domains and often struggle with fidelity. In contrast, we show that conceptually simple autoregressive video generation models based on a three-dimensional self-attention mechanism achieve competitive results across multiple metrics on popular benchmark datasets, for which they produce continuations of high fidelity and realism. We also present results from training our models on Kinetics, a large scale action recognition dataset comprised of YouTube videos exhibiting phenomena such as camera movement, complex object interactions and diverse human movement. While modeling these phenomena consistently remains elusive, we hope that our results, which include occasional realistic continuations encourage further research on comparatively complex, large scale datasets such as Kinetics.

## 1 Introduction

Generative modeling of video holds promise for applications such as content creation, forecasting, transfer learning and model-based reinforcement learning (Srivastava et al., 2015; Carl Vondrick, 2016; Oh et al., 2015; Kaiser et al., 2019). While recently there has been a lot of progress on generative models for text, audio and images, video generation remains challenging. To some extent this is simply due to the large amount of data that needs to be produced. Autoregressive models suffer from this particularly in their generation speed. On the other hand, they have a number of desirable attributes, such as their conceptual simplicity and tractable likelihood, which enables straightforward evaluation of their ability to model the entire data distribution.

Moreover, recent results on image generation by Menick & Kalchbrenner (2019) show that pixel-level autoregressive models are capable of generating images with high fidelity. These findings motivate the question of how far one can push such autoregressive models in the more general task of video generation when scaling recent advances in neural architectures to modern hardware accelerators.

In this work, we introduce a generalization of the Transformer architecture of Vaswani et al. (2017) using three-dimensional, block-local self-attention. In contrast to the block-local attention mechanism of Parmar et al. (2018), our formulation can be implemented efficiently on Tensor Processing Units, or TPUs (Jouppi et al., 2017). To further reduce the memory footprint, we combine this with a three-dimensional generalization of methods from Menick & Kalchbrenner (2019), who generate images as sequences of smaller, sub-scaled image slices.

Together, these techniques allow us to efficiently model videos as 3D volumes instead of sequences of still image frames, with direct interactions between representations of pixels across the spatial and temporal dimensions.

---

[*]Equal contribution.
[†]Work done while at Google Research.

We obtain strong results on popular benchmarks (Section 4.2, Appendix A) and produce high fidelity video continuations on the BAIR robot pushing dataset (Ebert et al., 2017) exhibiting plausible object interactions. Furthermore, our model achieves an almost 50% reduction in perplexity compared to prior work on autoregressive models on another robot pushing dataset.

Finally, we apply our models to down-sampled videos from the Kinetics-600 dataset (Carreira et al., 2018) (Section 4.3). While modeling the full range of Kinetics-600 videos still poses a major challenge, we see encouraging video continuations for a more limited subset, namely cooking videos. These feature camera movement, complex object interactions and still cover diverse subjects.

We hope that these initial results will encourage future video generation work to evaluate models on more challenging datasets such as Kinetics.

## 2 RELATED WORK

Our setup is closely related to that of Kalchbrenner et al. (2016), who extend work on pixel-level autoregressive image generation (van den Oord et al., 2016b;a) to videos. However, whereas they model the temporal and spatial dimensions separately with dilated convolutions and convolutional LSTMs, respectively, our model is conceptually simpler in that we do not make any distinction between temporal and spatial dimensions and instead rely almost entirely on multi-head self-attention (Vaswani et al., 2017) within the 3D video volume. For comparability, we provide results on Moving MNIST and another robot pushing dataset (Finn et al., 2016a) on which our model achieves an almost 50% reduction in perplexity (see Appendix A).

One major drawback of autoregressive models is their notoriously slow generation speed. However, we believe that further research into (partially) parallelizing sampling (Stern et al., 2018) and future hardware accelerators will help alleviate this issue and eventually make autoregressive modeling a viable solution even for extremely high-dimensional data such as videos.

To reduce the generally quadratic space complexity of the self-attention mechanism, we use block-local self-attention, generalizing the image generation approaches of Parmar et al. (2018) and Chen et al. (2018) to 3D volumes. In concurrent work, Child et al. (2019) instead use sparse attention after linearizing images to a sequence of pixels.

To further reduce memory requirements, we generalize sub-scaling (Menick & Kalchbrenner, 2019) to video. An alternative approach is optionally hierarchical multi-scale generation, which has recently been explored for both image generation (Reed et al., 2017; De Fauw et al., 2019) as well as video generation (Mathieu et al., 2016).

Earlier work on video generation mostly focused on deterministic approaches (Srivastava et al., 2015; Carl Vondrick, 2016; Xingjian et al., 2015; Liu et al., 2017; Jia et al., 2016), which fail to capture the high degree of stochasticity inherent in video. In response, a popular research direction has been that of generative latent-variable video models. In contrast to pixel-level autoregressive models, these posit an underlying latent process in tandem with the observed pixel values. Work in this category includes variants of variational autoencoders (Babaeizadeh et al., 2018; Denton & Fergus, 2018). To address the issues inherent in these models, most notably the tendency to generate blurry outputs possibly due to restricted modeling power, inadequate prior distributions, or optimization of a lower bound in place of the true likelihood, various directions have been explored, including the use of adversarial objectives (Mathieu et al., 2016; Vondrick et al., 2016; Lee et al., 2018), hierarchical latent-variables (Castrejón et al., 2019), or flow-based models (Kumar et al., 2019). All of these approaches admit significantly faster generation. However, in the adversarial case, they tend to only focus on a subset of the modes in the empirical distribution while flow-based models struggle with limited modeling power even when using a large number of layers and parameters.

A large fraction of earlier work on video generation has encoded specific intuitions about videos, such as explicit modeling of motion (Finn et al., 2016b; Denton & Fergus, 2018) or generation of optical flow (Pătrăucean et al., 2016). The conceptual simplicity of our model, however, is more in line with recent approaches to video classification that process videos by means of 3D convolutions (Carreira & Zisserman, 2017; Xie et al., 2018) or, similar to this work, spatiotemporal self-attention (Girdhar et al., 2018).

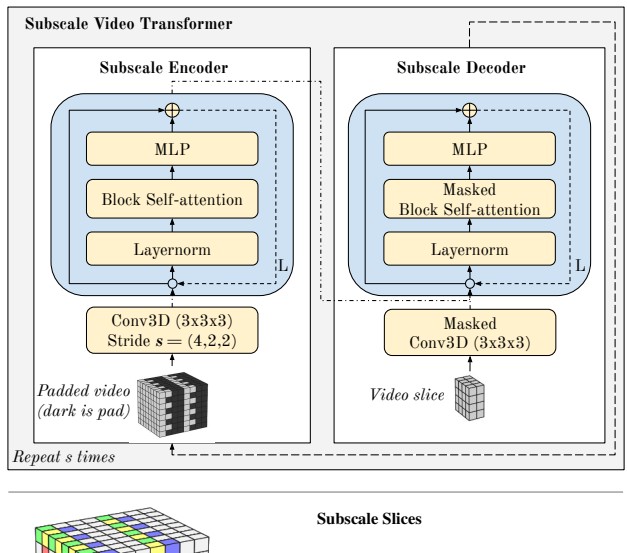

Figure 1: **Top**: Illustration of the subscale video transformer architecture and process flow. We incrementally generate $s = 4 \cdot 2 \cdot 2 = 16$ video slices. The video slices and their respective generation order are derived from subscaling. In each iteration, we first process the partially padded video (illustrated for slice index $(1, 0, 1)$, black means padding and gray means already generated or visible) by an encoder, the output of which is used as conditioning for decoding the current video slice. After generating a slice we replace the respective padding in the video with the generated output and repeat the process for the next slice. **Bottom**: Subscaling in 3D (best viewed in color). The 3D volume is evenly divided by a given subscale factor, here $s = (4, 2, 2)$, and the respective slices are extracted. The whole volume is generated by incrementally predicting the individual, much smaller slices, starting at slice $\boldsymbol{x}_{(0,0,0)}$ (yellow), followed by $\boldsymbol{x}_{(0,0,1)}$ (green), $\boldsymbol{x}_{(0,1,0)}$ (red), etc., in raster-scan order.

## 3 VIDEO TRANSFORMER

We generalize the one-dimensional Transformer (Vaswani et al., 2017) to explicitly model videos represented as three-dimensional spatiotemporal volumes, without resorting to sequential linearization of the positions in the volume (Child et al., 2019). This allows for maintaining spatial neighborhoods around positions, which is important as the large number of individual positions to be predicted in a video requires limiting the receptive field of the self-attention mechanism to a neighborhood around every position to avoid the quadratic blow-up in memory consumption of naive fully-connected attention.

We model the distribution $p(\boldsymbol{x})$ over videos $\boldsymbol{x} \in \mathbb{R}^{T \times H \times W \times N_c}$ — with time, height, width and channel dimensions, respectively — by means of a pixel-channel level autoregressive factorization.[1] That is, the joint distribution over pixels is factorized into a product of channel intensities for all $N_c$ channels, for each of the $N_p = T \cdot H \cdot W$ pixels, with respect to an ordering $\pi$ over pixels:

$$p(\boldsymbol{x}) = \prod_{i=0}^{N_p-1} \prod_{k=0}^{N_c-1} p(\boldsymbol{x}_{\pi(i)}^k | \boldsymbol{x}_{\pi(<i)}, \boldsymbol{x}_{\pi(i)}^{<k}). \tag{1}$$

The ordering $\pi$ is given by a combination of a subscale- and raster-scan ordering, as detailed in 3.2.

### 3.1 BLOCK-LOCAL SELF-ATTENTION

The attention mechanism of the original Transformer lets each element in a set of $N_p$ elements connect to every other element, via the fully-connected weighted adjacency (attention) matrix $A \in \mathbb{R}^{N_p \times N_p}$, with $A_{ij}$ representing attention weights from element $i$ to element $j$. Because $A$ grows quadratically with the number of elements it becomes prohibitively large for objects such as videos, which typically consist of hundreds of thousands of pixels or more. Therefore, similar in spirit to Parmar et al. (2018), we propose to use local self-attention by dividing a video into much smaller non-overlapping sub-volumes, or *3D blocks*. We then apply self-attention separately within each block. This approach is conceptually simple and amenable to highly efficient implementation on

---

[1]In the following, we denote general tensors by boldface lowercase letters and matrices by capital letters.

TPUs, which enables us to scale our models substantially while maintaining a comparatively high training speed with only a modest sacrifice in expressive power.

The Video Transformer consists of multiple stacked self-attention layers. Each layer divides the overall video volume of shape $(T, H, W)$ into smaller blocks of shape $(t, h, w)$ of length $n_p = t \cdot h \cdot w$, and performs attention within each block independently. Given a (flattened) block representation $\boldsymbol{z} \in \mathbb{R}^{n_p \times d}$ of hidden size $d$ as input, this amounts to:

$$[\boldsymbol{q}, \boldsymbol{k}, \boldsymbol{v}] = \text{layernorm}(\boldsymbol{z})W_{qkv} \qquad \boldsymbol{q}, \boldsymbol{k}, \boldsymbol{v} \in \mathbb{R}^{n_p \times d_a}, W_{qkv} \in \mathbb{R}^{d \times 3d_a}, \quad (2)$$

$$A = \text{softmax}\left(\boldsymbol{q}\boldsymbol{k}^\top / \sqrt{d_a} + B\right) \qquad\qquad A, B \in \mathbb{R}^{n_p \times n_p}, \quad (3)$$

$$\text{attention}(\boldsymbol{z}) = A\boldsymbol{v}. \qquad\qquad\qquad\qquad\qquad\qquad\qquad (4)$$

The input is first projected to query, key and value representations (Eq. 2). The attention matrix $A$ is then formed as the scaled dot-product between all query-key pairs adding a relative position bias $B$ (Parikh et al., 2016) (Eq. 3). The bias $B_{ij}$ is defined as the sum of per-dimension relative distance biases between element $i$ and $j$, along each of the time- and spatial dimensions. Finally, the values are aggregated with respect to the attention weights (Eq. 4).

Following Vaswani et al. (2017), we concatenate the output of $n_a$ parallel attention heads in each layer and project the result by a linear transformation (Eq. 5) before applying a residual connection. Finally, the output of the multi-head self-attention layer is passed through another dense layer with ReLU activation, followed by a final linear transformation and a residual connection (Eq. 6):

$$\tilde{\boldsymbol{z}} = [\text{attention}_1(\boldsymbol{z}); \cdots ; \text{attention}_{n_a}(\boldsymbol{z})] W_p + \boldsymbol{z} \qquad W_p \in \mathbb{R}^{(n_a \cdot d_a) \times d}, \quad (5)$$

$$\boldsymbol{z}' = \text{relu}(\text{layernorm}(\tilde{\boldsymbol{z}}) T_1) T_2 + \tilde{\boldsymbol{z}} \qquad\qquad T_1, T_2 \in \mathbb{R}^{d \times d}, \quad (6)$$

where overloading notation, $\text{attention}(\boldsymbol{z})$ denotes the blockwise application of self-attention to $\boldsymbol{z}$. Similar to Baevski & Auli (2019), we found that applying layer normalization before each block, rather than after each block as proposed by Vaswani et al. (2017), improves training.

**Connectivity.** Operating on 3D sub-volumes (blocks) of videos means that there is no direct information exchange between blocks. However, this can be addressed by varying the block sizes between each layer. To achieve this, we define blocks that stretch over the entire extent of at least a single dimension in each layer. Following this procedure, we can effectively connect all pixel positions in the encoder, but due to masking some dependencies are missed in the decoder. However, in our experiments these did not produce any visible, systematic artifacts. We discuss missing dependencies and potential remedies in Appendix C.

**Efficiency.** Running block-local self-attention is very efficient in practice as the cost of splitting videos into blocks is negligible. The approach of Parmar et al. (2018) uses overlapping 2D image blocks in each layer. We found this prohibitive as the required data copying is comparatively expensive. To avoid the need for overlaps to connect pixels across blocks, we simply vary block sizes between layers, which is highly efficient and, as our results show, works well in practice.

## 3.2 SPATIOTEMPORAL SUBSCALING

Menick & Kalchbrenner (2019) recently proposed generating images as a sequence of subscaled image slices. We similarly define a *subscale factor* $\boldsymbol{s} = (s_t, s_h, s_w)$ which divides a video into $s = (s_t \cdot s_h \cdot s_w)$ sub-sampled videos (*slices*), each of resolution $(T/s_t, H/s_h, W/s_w)$, as depicted in the bottom part of Figure 1. The slices are generated in order according to their respective offsets, such that we first generate slice $\boldsymbol{x}_{(0,0,0)}$, then $\boldsymbol{x}_{(0,0,1)}$, up until slice $\boldsymbol{x}_{(s_t-1,s_h-1,s_w-1)}$. Generating all slices one at a time in this way drastically reduces the number of pixels in memory to $N_p/s$, which enables scaling our architectures by a factor of $s$. Each slice is internally generated according to the raster-scan order. In the following we explain how slices are generated and how they are conditioned on already decoded slices. An overview is illustrated in the upper part of Figure 1.

**Slice Encoder.** The current slice $\boldsymbol{x}_{(a,b,c)}$ is generated conditioned on the encoded pixels from preceding slices as follows. First, we create a partially masked video, where only the pixels of preceding slices $\boldsymbol{x}_{<(a,b,c)}$ are visible. The partially masked video is then embedded by concatenating the one-hot encoding of the discretized pixel intensities of each channel. Subsequently, a 3D convolution with kernel size $\boldsymbol{k} = (k_1, k_2, k_3)$ and stride $\boldsymbol{s}$ (the sub-scaling factor) results in an encoded video

of resolution $(T/s_t, H/s_h, W/s_w)$. We apply convolution padding depending on the current slice index $(a, b, c)$. In particular, we pad with $(\lfloor k_1/2 \rfloor - a, \lfloor k_2/2 \rfloor - b, \lfloor k_3/2 \rfloor - c)$, which "centers" the convolution kernel on the pixels of the current slice. Finally, we add positional embeddings for each axis, as well as embeddings for the current slice index $(a, b, c)$, to the output of this strided convolution. The result is an initial encoder representation $\boldsymbol{z}^0_{(a,b,c)} \in \mathbb{R}^{T/s_t \times H/s_h \times W/s_w \times d_e}$, where $d_e$ is the embedding size. We can optionally condition on auxiliary information, such as per-frame action values of a robot arm, by concatenating this information to the initial encoder representation.

This representation is further transformed by a linear projection to hidden size $d$, before being fed as input to a stack of $L$ block-local self-attention layers as described in §3.1. Each layer is parameterized by a different block size and number of attention heads. The resulting output $\boldsymbol{z}^L_{(a,b,c)}$ is used as conditional input to the subscale slice decoder, which generates the pixels of the current slice $(a, b, c)$.

**Slice Decoder.** The pixel values of the current slice $\boldsymbol{x}_{(a,b,c)}$ are predicted conditioned on the encoder representation $\boldsymbol{z}^L_{(a,b,c)}$. The decoder is almost identical to the encoder in structure, except for the use of masking in the decoder as defined by the generation order. First, we embed $\boldsymbol{x}_{(a,b,c)}$ by summing $N_c$ channel embeddings of size $d_e$ at every pixel, before applying a 3x3x3 masked convolution (van den Oord et al., 2016a) on the embedded pixels, effectively representing each pixel by its already generated, immediate neighbors. Similar to the encoder, we add positional embeddings for the space- and time dimensions to the output of this masked convolution. As in the encoder, this results in an initial decoder representation $\boldsymbol{y}^0_{(a,b,c)} \in \mathbb{R}^{T/s_t \times H/s_h \times W/s_w \times d}$.

To condition on the encoder state, a linear projection of $\boldsymbol{z}^L_{(a,b,c)}$ is added to $\boldsymbol{y}^0_{(a,b,c)}$ and the resulting representation is fed through a stack of $L$ block-local self-attention layers, with masking, to produce a state $\boldsymbol{y}^L_{(a,b,c)}$ on which the final channel predictions are conditioned.

### 3.3 CHANNEL PREDICTION & LOSS FUNCTION.

The per-pixel channel intensities $\boldsymbol{x}^k_{(a,b,c)}$ (we omit the slice index $(a, b, c)$ in the following) for each channel $k < N_c$ are predicted by MLPs with a single hidden layer (Eq. 8), conditioned on the flattened final decoder state $\boldsymbol{y}^L \in \mathbb{R}^{n_p \times d}$ — which is itself conditioned on $\boldsymbol{z}^L_{(a,b,c)}$ and hence on prior slices $\boldsymbol{x}_{<(a,b,c)}$ — as well as the preceding channels $(\boldsymbol{x}^j)_{j=1\ldots k-1}$ for each pixel, encoded as one-hot vectors. Finally, the per video slice loss is defined as the negative log-likelihood as in Eq. 9:

$$\boldsymbol{u}^k = \left[ \text{layernorm}\left(\boldsymbol{y}^L\right); \text{onehot}\left(\boldsymbol{x}^1\right); \cdots; \text{onehot}\left(\boldsymbol{x}^{k-1}\right) \right] U_k, \tag{7}$$

$$p\left(x^k_i | \boldsymbol{x}^{<k}_i, \boldsymbol{x}_{<i}\right) = \text{softmax}\left(\text{relu}(\boldsymbol{u}^k_i)P\right), \qquad P \in \mathbb{R}^{d \times N_v}, \quad U_k \in \mathbb{R}^{(d+(k-1)\cdot N_v) \times d}, \tag{8}$$

$$\mathcal{L}(\boldsymbol{x}) = -\sum_{i=0}^{n_p-1} \sum_{k=0}^{N_c-1} \ln p(x^k_i | \boldsymbol{x}^{<k}_i, \boldsymbol{x}_{<i}). \tag{9}$$

We found that splitting the color channel values of the videos into coarse and fine bits helps slightly in terms of performance. Specifically, we split the $3 \times 8$-bit RGB channels into $6 \times 4$-bit channels ($N_c = 6$, $N_v = 16$), such that the coarse bits of all three channels are predicted before the fine bits. Furthermore, splitting channels this way at the input level considerably lowers memory footprint when encoding videos as onehot vectors on TPUs.

## 4 EXPERIMENTS

Below, we provide details on the model variants considered, our training setup and the evaluation metrics used. We focus our evaluation on the BAIR Robot Pushing and Kinetics datasets. Additional results on Moving MNIST and another robot pushing dataset are provided in Appendix A for reference. Sample videos strips of each model and dataset can be found in Appendix F and sample videos at `https://bit.ly/2Zb017f`.

## 4.1 MODELS & SETUP

Unless specified otherwise, we model video slices of 4 frames with a spatial resolution of 32x32. Both the encoder and decoder consist of 8 layers and have a nearly identical structure, except for the use of masking in the decoder, as described in Section 3.2. We apply block-local self-attention with the following block sizes $(t, h, w)$. Layers 1-4: (4, 8, 4); (4, 4, 8); (1, 32, 4); and (1, 4, 32). Intuitively, layers 1 and 2 are responsible for gathering temporal information whereas layers 3 and 4 gather spatial information of the entire frame. Layer 3 has access to the entire height and layer 4 to the entire width of a frame. The remaining 4 layers have the same block sizes, but in reverse order. However, as discussed in Appendix B, this particular choice of block size ordering is not crucial. There are $n_a = 8$ attention heads, each with hidden size $d_a = 128$. Our base models are trained with embedding size $d_e = 128$ and hidden size of $d = 512$ (46M parameters). Based on ablations in Appendix B, we observed that increasing the hidden dimension is preferable to using deeper networks. Hence, we increase the hidden size to $d = 2048$ and use $n_a = 16$ instead of 8 heads for the last 4 encoder/decoder layers in our large models (373M parameters).

**Models.** To assess the effect of subscaling, we explore the following variants. These differ mainly in the subscaling factor $s$ as well as the context kernel size $k$, defaulting to $k = s$:

***Spatiotemporal Subscaling.*** The subscale video transformer with full spatiotemporal subscaling applies subscaling in every dimension. For instance, a 16x64x64 video is subscaled by factors $s = (4, 2, 2)$ to 16 slices of 4x32x32.

***Spatial Subscaling.*** This model uses no temporal subscaling and only subscales individual frames to a resolution of 32x32. For instance, a 4x64x64 video is subscaled by factors $s = (1, 2, 2)$ to 4 slices of 4x32x32.

***Single Frame.*** This model uses no subscaling. Instead, we here model an entire single frame at a time, conditioned only on the previous three frames to limit memory consumption. The model uses no actual subscaling. Instead, one can imagine a 16x64x64 video to be subscaled by factors $s = (16, 1, 1)$ to 16 slices of 1x64x64 frames. The context kernel size is $k = (6, 1, 1)$ which means that we merely condition on a context of 3 past frames, as the current and future frames are always masked when the temporal subscaling factor equals the full video length. Self-attention blocks are adapted as follows: Layers 1-4: (1, 8, 16); (1, 16, 8); (1, 2, 64); (1, 64, 2). For the remaining 4 layers we use the same blocks, again in reverse order.

**Training.** All models are trained with RMSProp (Tieleman & Hinton, 2012) with a fixed learning rate of $2 \cdot 10^{-5}$, decay of $0.95$ and momentum of $0.9$. We use a batch size of 64 video slices, if not stated otherwise, and shuffle the slices to avoid having all slices in a batch correspond to the same video. The smaller models are trained for 300K steps and the larger ones for 1M steps. No explicit regularization is applied as we could not observe any form of over-fitting. Videos longer than the training resolution are cropped randomly in time to the defined training length. If not stated otherwise, models are conditioned on the first frame during training, which is achieved by masking the loss corresponding to this frame. In preliminary experiments, this gave a minor improvement over computing the training loss across all frames.

**Intrinsic Evaluation.** Most results are reported as bits per dimension (bits/dim), the average negative $\log_2$-probability assigned by the model per (RGB) channel, averaged across all pixels in the video. This corresponds directly to the loss optimized by the model. In all experiments, we condition (prime) on a specified number of initial frames. The log-probabilities corresponding to these frames are excluded from this average.

**Extrinsic Evaluation.** Prior work mainly reported results on the peak signal-to-noise ratio (PSNR) and mean-structural similarity (SSIM) metrics (Wang et al., 2004b). However, these metrics were developed for images and have serious flaws when applied to videos (Wang et al., 2004a; Wang & Li, 2007; Zhang et al., 2018; Lee et al., 2018). Conceptually, PSNR has a strong preference for blurry videos as it is based on pixel-level mean squared error. Similarly, SSIM does not correlate well with perceptual quality either. For instance, variational autoencoders show very strong performance on this metric despite producing blurry videos (Lee et al., 2018). Hence, we focus on the Fréchet Video Distance (FVD), which was recently proposed by Unterthiner et al. (2018) as a qualitative metric sensitive to visual quality, temporal coherence and diversity of samples. This is the spatiotemporal counterpart to the Fréchet Inception Distance (Heusel et al., 2017), replacing the ImageNet-trained

Table 1: Quantitative results on BAIR Robot Pushing (left) and Kinetics (right).

| Models | Bits/dim | FVD | FVD (Avg) |
|---|---|---|---|
| Single Frame | 1.49 | 104±4 | 99±2 |
| Spatial Sub. | 1.57 | 111±4 | 108±1 |
| Spatiotemp. Sub. | 1.53 | 106±3 | 106±2 |
| Spatiotemp. Sub. (L) | **1.35** | **94±2** | **96±2** |
| SV2P [1]† | – | 263‡ | – |
| SAVP [2]† | – | 116‡ | – |
| VideoFlow [3] | 1.87‡ | – | – |

(a) **BAIR Robot Pushing.** Bits/dim averaged across 15 subsequent frames when priming with 1 initial frame, FVD and unrolled average FVD scores. Best results in bold. † Results from Unterthiner et al. (2018). ‡ Results are not strictly comparable (see text for details). [1] Babaeizadeh et al. (2018), [2] Lee et al. (2018), [3] Kumar et al. (2019).

| Models | Bits/dim | FVD | FVD (Avg) |
|---|---|---|---|
| Single Frame | 1.40 | 243±6 | 413±11 |
| Spatial Sub. | 1.47 | 263±6 | 450±15 |
| Spatiotemp. Sub. | 1.49 | 195±7 | 375±11 |
| Single frame (L) | **1.14** | 207±8 | 353±13 |
| Spatiotemp. Sub. (L) | 1.19 | **170±5** | **316±12** |

(b) **Kinetics.** Bits/dim averaged across 15 subsequent frames when priming with 1 initial frame, FVD and unrolled average FVD scores when priming with 5 frames. Best results in bold.

Inception network of the latter with an I3D Network trained on Kinetics. Despite sharing the known drawbacks of FID (Bińkowski et al., 2018), FVD has shown to correlate much stronger with human raters compared to both PSNR and SSIM (Unterthiner et al., 2018). We report the FVD of the first 16 frames, as well as the "unrolled" average FVD across all contiguous subsequences of 16 frames. In each case, we report the mean and standard deviation of 20 trials.

**Sampling time.** Sampling from autoregressive models is notoriously slow. However, because our decoders are not very deep (8 layers) we are able to sample a batch of four 30x64x64 videos in acceptable time (approx. 8 minutes) with our large models on a Nvidia Tesla V100. Though this might still be impractical we argue that further advances in parallel sampling strategies (Stern et al., 2018) and future hardware will alleviate this disadvantage significantly.

## 4.2 BAIR ROBOT PUSHING

BAIR Robot Pushing (Ebert et al., 2017) shows a robotic arm pushing and grasping objects in a box. It consists of roughly 40K training- and 256 test videos. We prime on the first frame for training and evaluation.

**Empirical Results.** All variants of the Video Transformer achieve strong results compared to prior work in terms of both intrinsic and extrinsic metrics. From Table 1a, we see that the small models already reduce the perplexity in terms of bits/dim by almost 20% compared to the recently proposed VideoFlow model (Kumar et al., 2019) with our large model (L) reducing perplexity even further to a 25% improvement. Similar to Menick & Kalchbrenner (2019), we find that subscaling can have a slightly negative effect on bits/dim. In terms of perceptual quality, every incarnation of our model obtains a lower (better) FVD score compared to all models evaluated by Unterthiner et al. (2018), which notably includes adversarial networks with no guarantees of covering the full empirical distribution. These results are not strictly comparable, since prior work has used longer priming sequences of two (Babaeizadeh et al., 2018; Lee et al., 2018) or three (Kumar et al., 2019) frames, whereas our models (to our disadvantage) see a single prime frame. Note that we sample with temperature 0.9 for the extrinsic metrics as we observed improved qualitative results at this temperature on the validation set. This corresponds to a mild form of mode dropping and is common practice to improve sampling quality. For fair comparison we also tweaked the "temperature" of SAVP by scaling the variance of its normal distribution when sampling. This, however, did not result in any improvements for FVD.

Further results on an earlier version of robot pushing (Finn et al., 2016a) and Moving MNIST (Srivastava et al., 2015) can be found in Appendix A for brevity. In summary, like Kalchbrenner et al. (2016), we match the lower bound on Moving MNIST while obtaining an *almost 50% reduction in bits/dim on robotic pushing* which demonstrates the superiority of our models against prior work on autoregressive video modeling.

**Qualitative Observations.** All variants of our model reach similar quantitative results on these benchmarks and we observe no immediate differences in fidelity. However, there are some notable differences. First, whereas the spatiotemporal subscaling model is able to capture temporal depen-

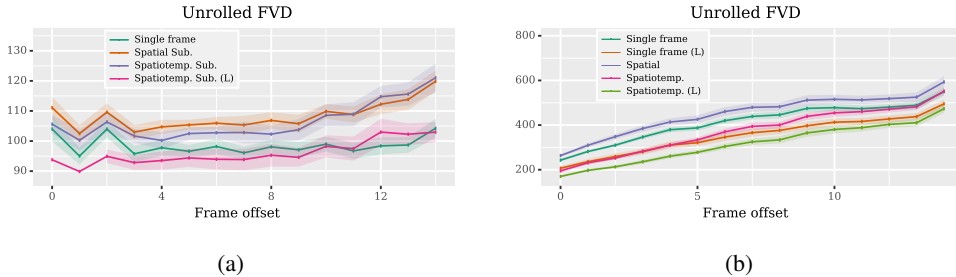

Figure 2: Unrolled FVD metrics on BAIR Robot Pushing (left) and Kinetics (right).

dencies across up to 16 frames (given subscaling in time by a factor four), the remaining models can only capture dependencies across four frames. This can, for example, result in deformation of occluded objects (e.g., Figure 4 of the Appendix). However, due to the simplicity of the benchmark datasets, this is not appropriately reflected in the metrics including better unrolled FVD curves for the single frame base model in Figure 2a. Second, we observe that lowering the sampling temperature from 1.0 to 0.9 consistently improves results. Notably, spatiotemporal subscaling seems more robust to sampling errors as its performance decays less when sampling with temperature 1.0 ($122\pm4$ Avg. FVD) compared to the spatial subscaling ($134\pm4$) and single frame models ($153\pm7$). We attribute this finding to the difference in generation order when spatiotemporal subscaling is employed as it predicts pixels over the entire extend of the 3D video volume early and thereby effectively anchors future predictions around these pixels. Finally, considering that our results on BAIR Robot Pushing in terms of FVD are on par with those between two ground-truth subsamples (Figure 4 of Unterthiner et al. (2018)), we may be approaching the limit of this benchmark. On the other hand, it could be that FVD suffers out-of-domain and is not sufficiently sensitive to long-range temporal dynamics, since it is trained to perform human action recognition, which is known to predominantly rely on local features (Carreira & Zisserman, 2017; Xie et al., 2018).

## 4.3 KINETICS

Moving from a constrained to a real world setting, we next apply our models to the Kinetics dataset (Kay et al., 2017), a large scale action-recognition dataset consisting of YouTube videos. Specifically, we use Kinetics-600, which contains roughly 400K training videos ranging over 600 action classes (Carreira et al., 2018). We center-crop and down-sample each frame to 64x64 with a width-3 Lanczos filter and anti-aliasing.

We introduce a slight change to our setup by using a separate decoder for the first slice $x_{(0,0,0)}$. This decoder can be twice as deep (16 instead of 8 layers) as the original subscale decoder, because it does not rely on any encoder. For all other slices we train a regular subscale model (8 layers in both encoder and decoder) as before. Using a separate first-slice decoder means that there is no wasted encoder computation on the first slice and that there are additional parameters. Furthermore, for our large models we scale the batch size to 256 by training in parallel on 128 TPU v3 instances for 1M steps.

**Empirical Results.** Results for our base models are shown in the upper part of Table 1b. In line with results on BAIR pushing, we find that the single frame model obtains better performance in terms of bits/dim. In contrast, we observe that the spatiotemporal subscaling model generates better and more robust video continuations which is reflected by its superior FVD scores. Our large models (L) show much stronger performance across the board (see lower half of Table 1b and Figures 2b), lowering the perplexity to 1.14 bits/dim for the single frame model. While the spatiotemporal subscaling model obtains slightly worse perplexity of 1.19 bits/dim, it improves FVD to 170. Despite its good performance on bits/dim, even with a temperature of 0.9, samples from the large single frame model are prone to instability and in many cases we observe color "explosions" (Figure 12 in the Appendix shows an example) which is reflected in its significantly higher FVD score. Although much less pronounced we observed such instability already when sampling with temperature 1.0 on BAIR pushing which clearly indicates the benefits of temporal subscaling for video generation.

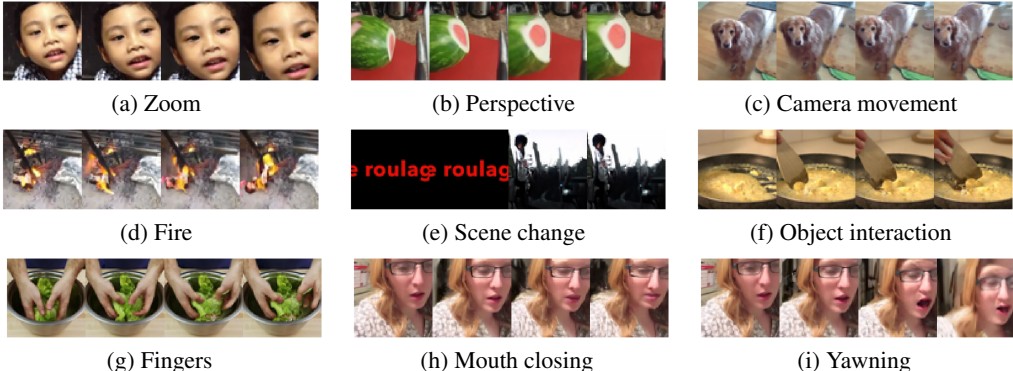

Figure 3: Selected Kinetics continuations from a set of 128 videos and 16 samples which showcase a variety of natural, video-specific phenomena our model learns to generate. We used our large spatiotemporal subscaling model and prime generation with 5 frames (0-4) to include the first two frames in subscale order (0, 4). Samples are generated with temperature of 0.9. The examples depict frames 0, 5, 10 and 15.

**Qualitative Observations.** Figure 3 shows samples from a cooking subset of Kinetics that we describe in Appendix E. These are selected to showcase different aspects of real-world videos learned by the large spatiotemporal subscaling model. Figures 3a and 3c demonstrate the model's ability to handle camera movement. We find that camera movement seems to be learned early in training, possibly since it is a major source of uncertainty. This requires transforming pixels correctly while hallucinating new pixels at the edges. Similarly, object movement resulting, for instance, in a change of perspective is predicted quite well (Figure 3b). Highly stochastic motion such as fire (Figure 3d) or steam is modeled surprisingly well. Videos in Kinetics sometimes contain scene changes and our model, too, occasionally generates videos with jumps to completely new scenes (Figure 3e). Motion of human fingers and faces seems challenging to model. Nevertheless, in a number of samples the model is able to generate somewhat believable continuations as can be seen in Figures 3g, 3h or 3i.

These selected examples show only a small subset of the interesting phenomena handled by the model and illustrate the sheer complexity involved in modeling this dataset. In Appendix F, we provide multiple samples, primed with the same initial frames to illustrate the diversity of the generated samples.

**Limitations.** While we obtained the occasional encouraging sampl, we would like to point out that the diversity of Kinetics still poses a major challenge. Failure modes range from freezing movement or object distortions to continuations that "wash out" entirely after a few frames. We firmly beleive that yet larger datasets and/or models will be required to capture the complexity of even short clips from YouTube videos. With this work we merely provide an initial baseline, hoping to highlight both the potential and the enormous room for improvement.

## 5  CONCLUSION

We presented an autoregressive model of videos based almost entirely on a variant of block-local self-attention that can easily be implemented efficiently on TPUs. Combined with spatiotemporal subscaling, our models can be scaled up substantially while retaining the ability to capture longer range spatiotemporal dependencies.

Empirically, we obtain state-of-the-art results across a range of video generation benchmarks, while the scalability of our approach enables us to make an initial attempt at modeling videos of unusually high complexity and diversity as found in the Kinetics dataset. Our models occasionally generate encouraging continuations, especially on a subset of cooking videos, yet we find modeling the full range of such videos clearly remains a major challenge.

## ACKNOWLEDGEMENTS

This work benefited from numerous conversations with Nal Kalchbrenner, as well as discussions with Jacob Menick, Mohammad Taghi Saffar and Niki Parmar. We would also like to thank Chelsea Finn and Tom Kwiatkowski for thoughtful comments on an earlier draft.

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

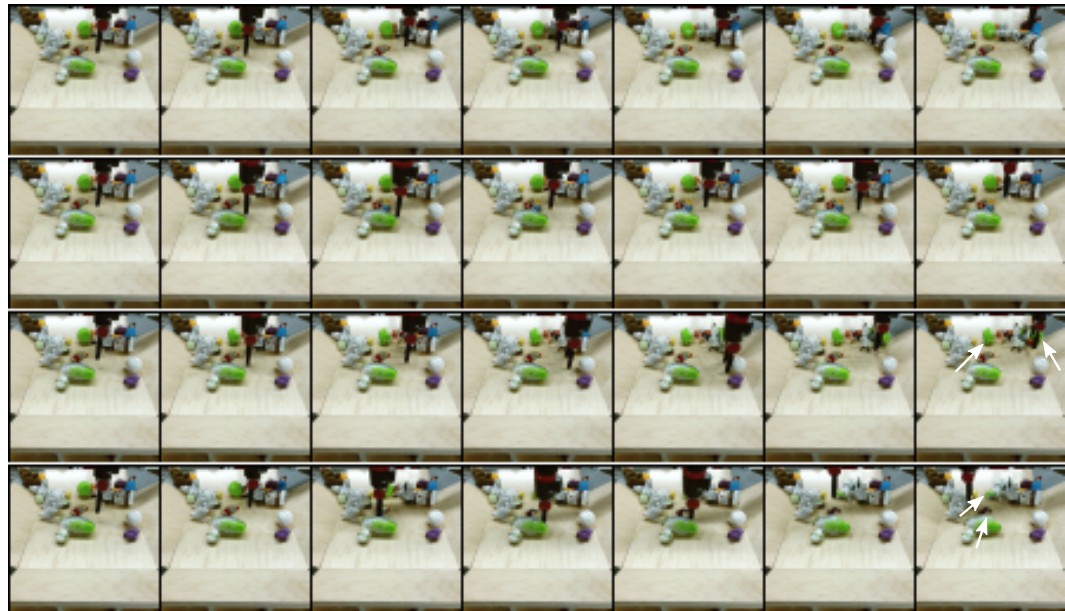

Figure 4: Samples (showing every 5th frame horizontally) illustrating occlusion effects on BAIR Robot Pushing. Models without temporal subscaling (rows 3-4) fail on occlusions, whereas the model with temporal subscaling (row 2) correctly maintains objects from the ground truth video (row 1). Notice the green ball deformation on rows 2 and 3 and the hallucinated green ball on the right edge of row 3, which are caused by missing temporal dependencies across the duration of occlusion.

Table 2: **Moving MNIST.** Nats per frame averaged across 10 subsequent frames when priming with 10 initial frames. Best results in bold. [†] The lower bound reported in (Kalchbrenner et al., 2016) is slightly higher than ours.

| **Models** | Nats/Frame ($\downarrow$) |
|---|---|
| Single Frame | **86.2** |
| Spatial Subscaling | 91.8 |
| Spatiotemporal Subscaling | 90.0 |
| VPN (Kalchbrenner et al., 2016) | 87.6 |
| Lower bound | 85.1 (86.3)[‡] |

## A  FURTHER BENCHMARKS

### A.1  MOVING MNIST

Moving MNIST (Srivastava et al., 2015) consists of 100K training- and 10K validation/test videos of two handwritten digits from the MNIST benchmark that move deterministically across the frame, crossing each other and bouncing off the borders. The partial occlusion of crossing digits makes this dataset challenging. To be comparable with Kalchbrenner et al. (2016), we use the first ten frames as priming and predict the subsequent ten frames.

To allow direct comparison with Kalchbrenner et al. (2016), we change our loss to a "deterministic" loss (and derived nats-per-frame metric) which is defined as: $H(z, y) = -\sum_i z_i \ln y_i + (1 - z_i) \ln(1 - y_i)$, where $z_i$ are the gray-scale targets between 0.0 and 1.0, and $y_i$ are the predicted scalar intensities.

From Table 2, we find that like Kalchbrenner et al. (2016) our single frame prediction model (i.e., no subscaling) virtually solves the task in the sense that it almost matches the lower bound of the

Table 3: Ablation of hyper-parameter settings in terms of bits per dimension for models on 256 BAIR Robot Pushing validation videos. All models were primed on 1 frame and trained for 300K steps with a batch size of 64.

| Layers | | Heads | | Hidden size | |
|---|---|---|---|---|---|
| 4 | 1.63 | 4 | 1.59 | 256 | 1.65 |
| 8 | 1.55 | 8 | 1.55 | 512 | 1.55 |
| 16 | 1.48 | 16 | 1.51 | 1024 | 1.47 |
| 24 | 1.45 | 24 | 1.47 | 2048 | **1.40** |

loss. However, this is not true for our subscaling models. Employing spatial subscaling on this task gives aliasing artifacts that make it harder to predict future frames. Although this finding is limited to Moving MNIST, it suggests that spatial subscaling can potentially hurt generation.

## A.2 ROBOTIC PUSHING.

Robotic Pushing (Finn et al., 2016a) was used in prior work on autoregressive video generation (Kalchbrenner et al., 2016). The videos show a robotic arm pushing and grasping objects in a box and there are roughly 50K training videos and 1500 test videos with seen and novel objects, respectively. Following prior work, we use the initial two frames for priming and condition on the robot arm action for each frame as described in Section 3.2. We use the same setup as (Kalchbrenner et al., 2016) with videos of twenty frames down-sampled to 64x64 with a Lanczos filter and anti-aliasing.

We report results to compare with prior work on autoregressive video generation by Kalchbrenner et al. (2016), who achieve 0.92 bits/dim (0.64 nats/dim) with 2 frames of priming on each of the test splits (one with objects seen during training and one with novel objects). We trained a large (2048 dimensional) spatiotemporal subscaling model which achieves 0.51 bits/dim on the subset with seen objects and 0.47 bits/dim on the subset with new objects, which corresponds to an **almost 50% reduction in perplexity**.

## B HYPER-PARAMETER SWEEPS

Table 3 shows the impact of different architectural settings. We see that the hidden size has the biggest impact followed by the number of layers and heads. This is an interesting as well as important finding because increasing the hidden size (wider networks) requires more parallel compute which modern Deep Learning hardware excels at. Computation time grows sub-linear, memory linear and parameters partially quadratically. In contrast all of these aspects grow linearly with deep networks. For scaling up architectures depth is therefore not the preferred option as we suffer much more in terms of computation time while having less parameters.

In another experiment, we shuffle the arrangement of block sizes between layers and found that it did not really matter, that is, all results were within 0.01 bits/dim. However, our setup had the best overall performance.

Finally, we tried sampling temperature 0.9 and 1.0 only on the BAIR Robot Pushing validation set and found that temperature 0.9 consistently gave more robust predictions and better results on all extrinsic metrics.

## C CONNECTIVITY IN BLOCK-LOCAL SELF-ATTENTION

**Blind Spots.** Varying block sizes between layers in block-local self-attention can efficiently connect every pixel with every other pixel when no masking is employed. If masking is employed to respect the generation order (as in our slice decoder) block-local self attention produces "blind spots" which leads to independence assumptions. To exemplify these special cases, consider position $(1, 0, 0)$, the top-left pixel of the second frame, and its direct predecessor in generation order $(0, h - 1, w - 1)$, the bottom-right pixel of the first frame. The only way to establish a connection

between these two positions is through a direct connection, because masking prevents any indirect connection. Thus, there has to be one layer in which both of these pixels are in the same block. This block must at least stretch over the entire extent of both width and height (i.e., the full frame) as well as at least 2 time steps. Running full self-attention in such blocks can easily become prohibitive for large $h$ and $w$.

**Remedies.** There seems to be no simple solution that solves the problem of blind spots completely. However, we can make sure that local dependencies up to a certain distance are all covered by increasing the kernel size of the initial, masked convolution in the decoder. It is also possible to combine block-local self-attention with its dual form, dilated self-attention in $n$ dimensions which connects all pixels at the same relative position within their respective block with each other. Finally, we find that it is important to avoid blocks of small sizes in any dimension (e.g., 1). That means, even if we stretch a block to the full extent of one dimension it is important to define sizes at least larger than 1 on all other dimensions to limit the number of unconnected pixels.

On the other hand, the independence assumptions due to masking do not seem to produce any systematic, visible artifacts in our samples. We believe this to be an interesting finding by itself as it shows that there is potential for parallelizing autoregressive video generation by systematically exploring further independence assumptions.

## D  ADDITIONAL FINDINGS

Below, we summarize some additional findings that may be of interest to some readers:

- We found that using blocks stretching across a single time-/row-/column- dimension, is substantially worse than using blocks that stretch at least to some extent in all directions. This is likely due to the fact that future masking in the decoder imposes strong independence assumptions in this case, as discussed in Appendix C.

- We found that RMSProp with momentum converges significantly faster than ADAM, which we tried with different learning rates and settings for $\beta_1$ and $\beta_2$.

- We tried using continuous, rather than discretized one-hot, input channel representations, but this had an overall negative impact on both performance and sample quality.

- We experimented with a gating mechanism in Eq. 3, such that the attention matrix $A$ is masked elementwise with $(1 - I)$ to allow for not attending to any element, similar to sentinel attention (Lu et al., 2017). However, this had no effect on generation quality.

## E  KINETICS COOKING

We found that for many video-prefixes in Kinetics it is very hard for our model to predict continuations. For instance, main objects in the videos are too small or movement is too fast which results in very blurry frames or there is little to no movement at all. Figure 13 shows some examples. Therefore, we created a subset of cooking videos that we found to exhibit these problems to a lesser degree.

In particular we filtered videos whose label matched the following regular expression:

```
.*(baking|barbequing|breading|cooking|cutting|pancake|vegetables|
   meat|cake|sandwich|pizza|sushi|tea|peeling|fruit|eggs|salad).*
```

Note that we still train on the full Kinetics training set and only use the cooking set to showcase samples in some cases.

## F  SAMPLES

Figures 5-8 show samples from our spatiotemporal subscaling and large spatiotemporal subscaling models on BAIR Robot Pushing. Figures 5 and 6 illustrate the fidelity and realism of the generated samples, whereas Figures 7 and 8 illustrate the diversity of samples.

Figures 9-11 show samples from our spatiotemporal subscaling model on cooking videos for Kinetics-600, while Figure 12 depicts samples from the single frame model. In each case, we prime on 5 frames and sample the next 11 frames. Each figure shows 16 different samples from the same model. As can be seen, the model is able to generate diverse continuations while retaining fidelity. For the single frame model we observe strange color artifacts (exploding colors) which we attribute to the standard, raster-scan generation order of this model.

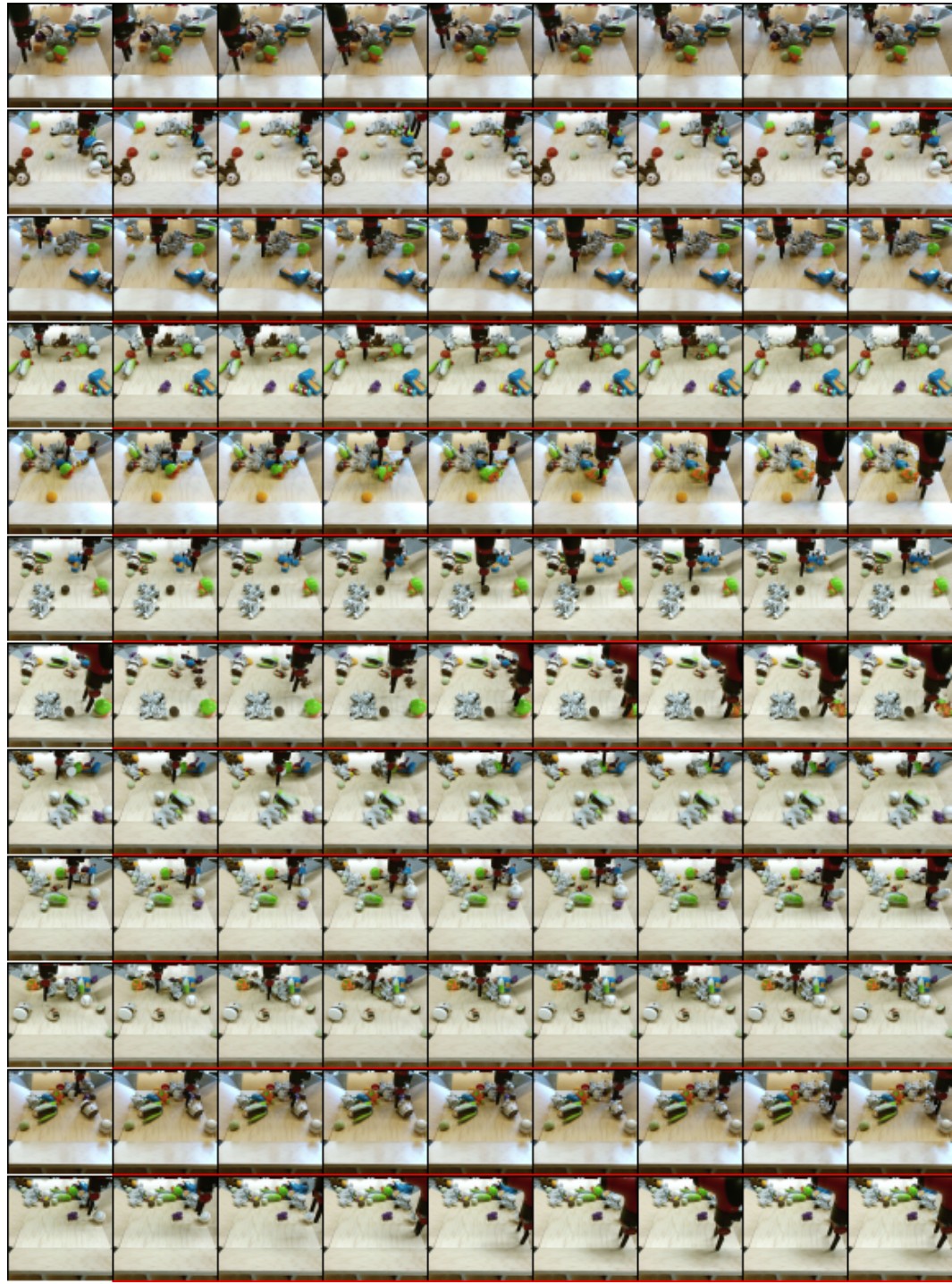

Figure 5: Samples of 30 future frames (showing every 4th frame) for 12 test videos with the spatiotemporal subscaling model, using 1 prime frame and temperature 0.9 on BAIR Robot Pushing.

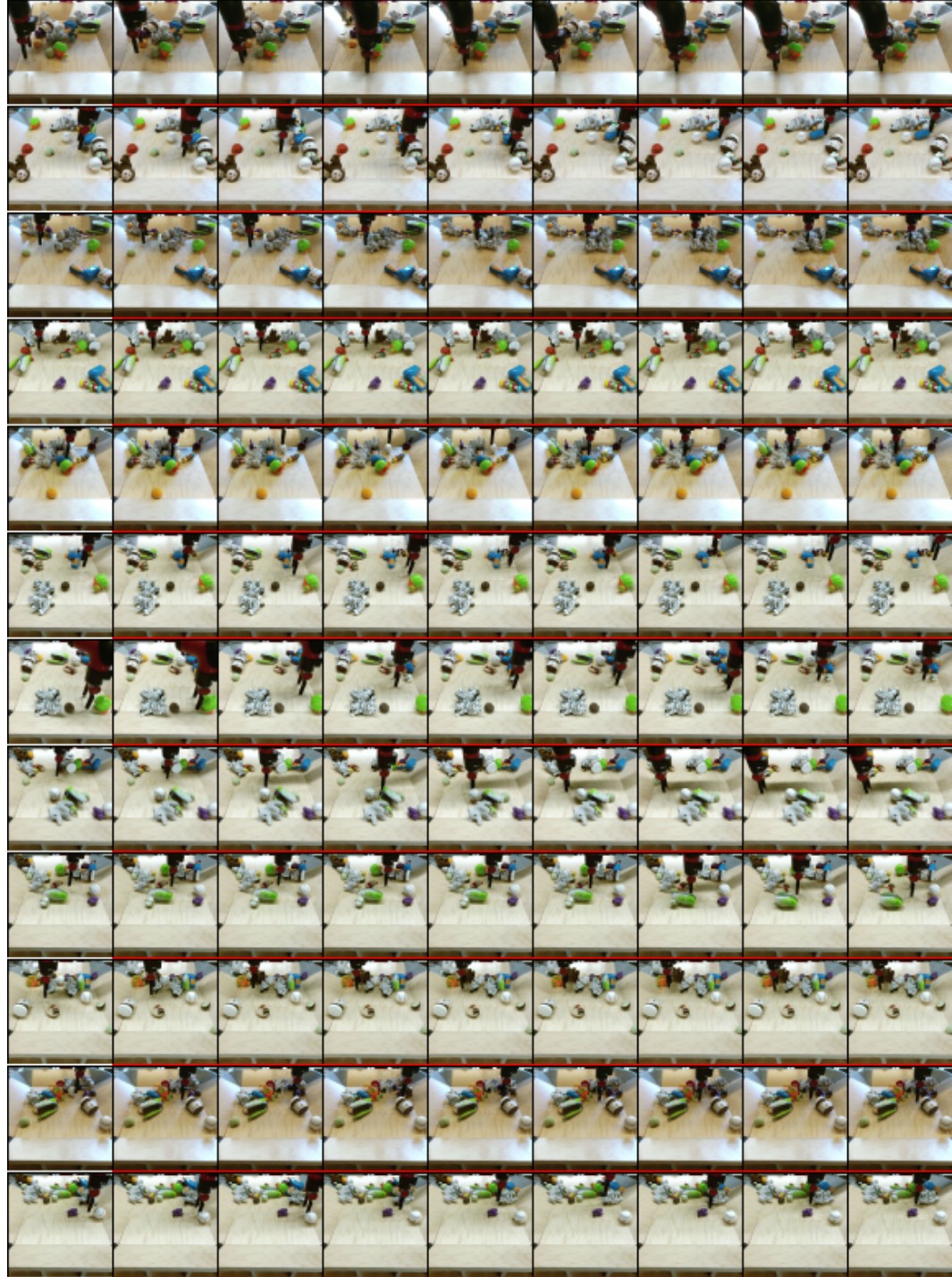

Figure 6: Samples of 30 future frames (showing every 4th frame) for 12 test videos with the large spatiotemporal subscaling model, using 1 prime frame and temperature 0.9 on BAIR Robot Pushing.

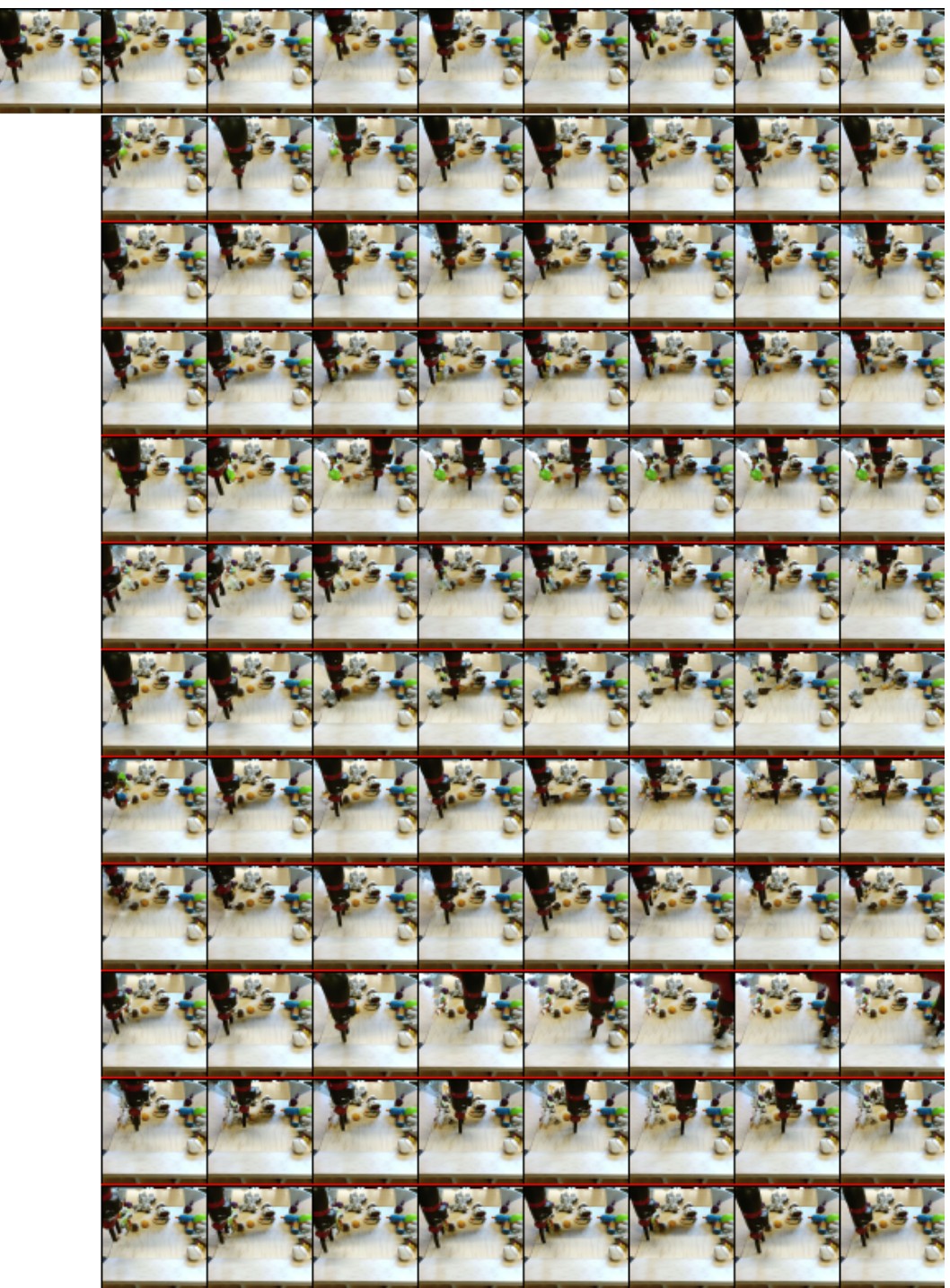

Figure 7: 11 samples of 30 future frames (showing every 4th frame) for 1 test video (top row) with the spatiotemporal subscaling model, using 1 prime frame and temperature 0.9 on BAIR Robot Pushing.

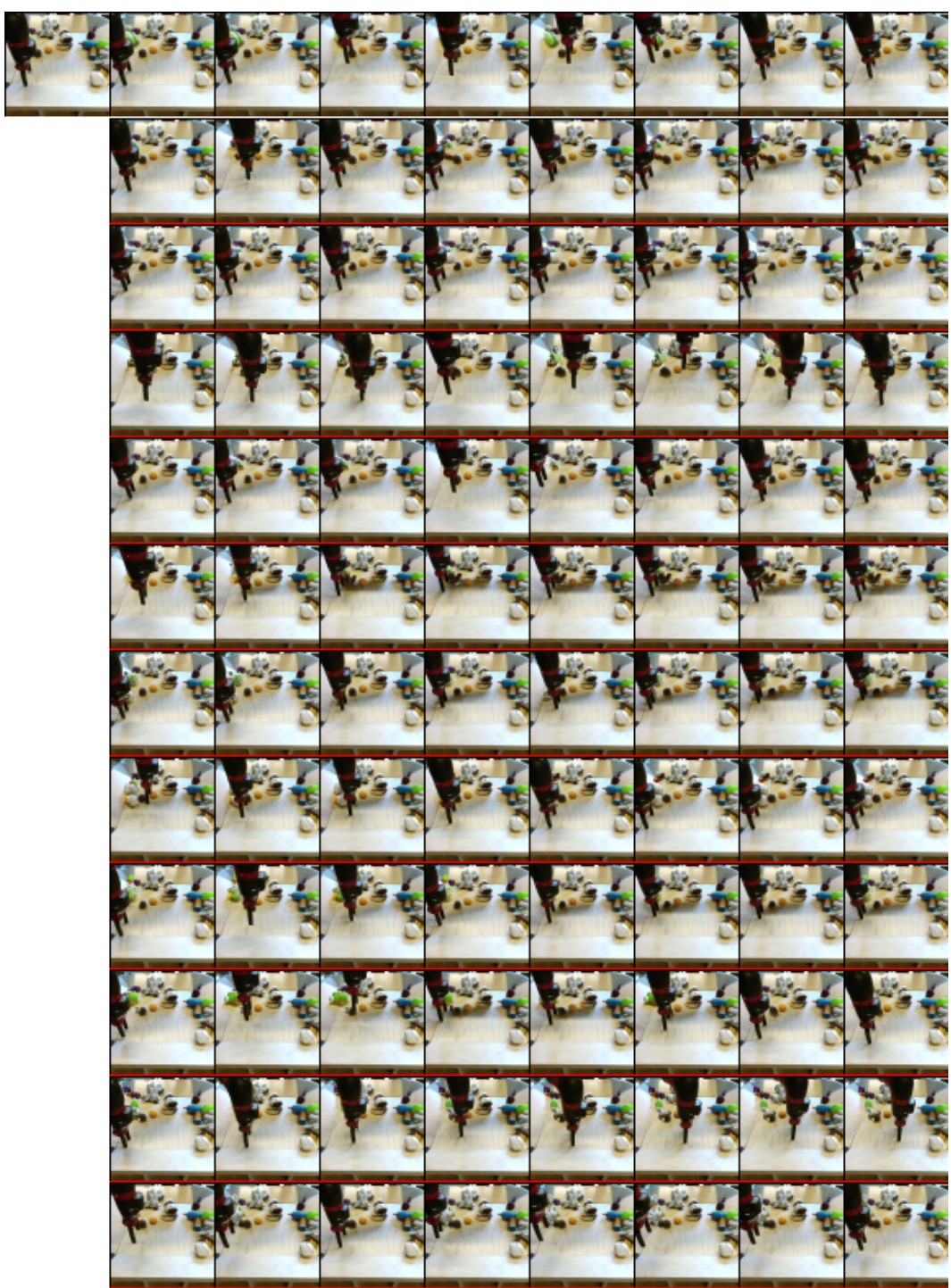

Figure 8: 11 samples of 30 future frames (showing every 4th frame) for 1 test video (top row) with the large spatiotemporal subscaling model, using 1 prime frame and temperature 0.9 on BAIR Robot Pushing.

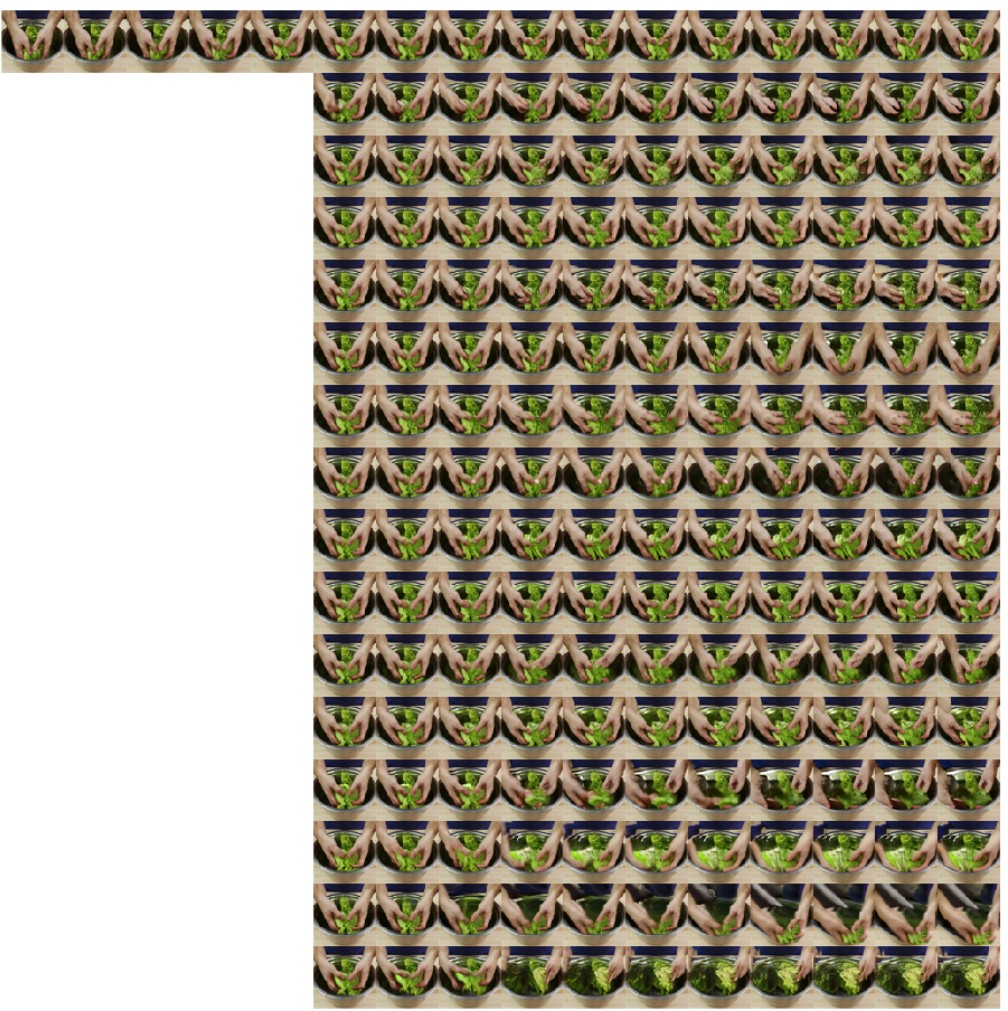

Figure 9: Samples of 11 future frames from the spatiotemporal subscaling model with 5 prime frames on 64x64 Kinetics.

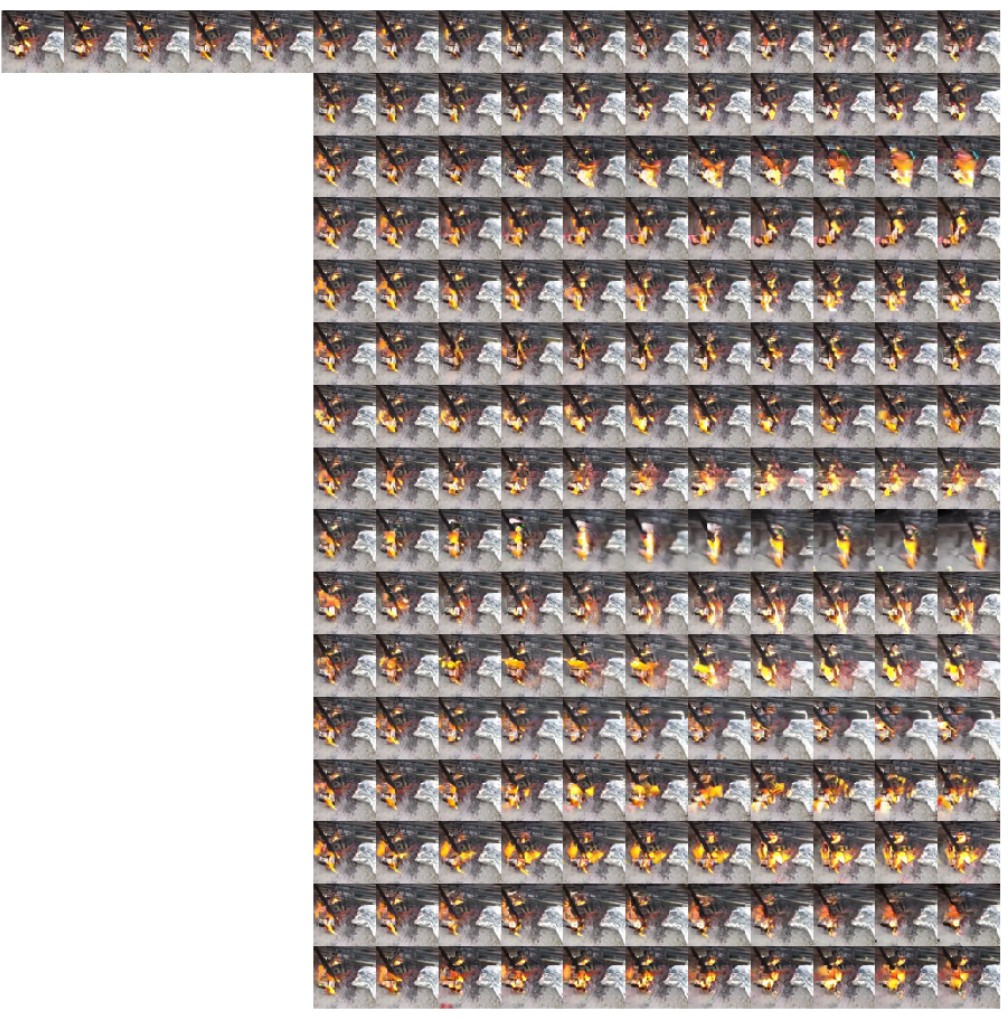

Figure 10: Samples of 11 future frames from the spatiotemporal subscaling model with 5 prime frames on 64x64 Kinetics.

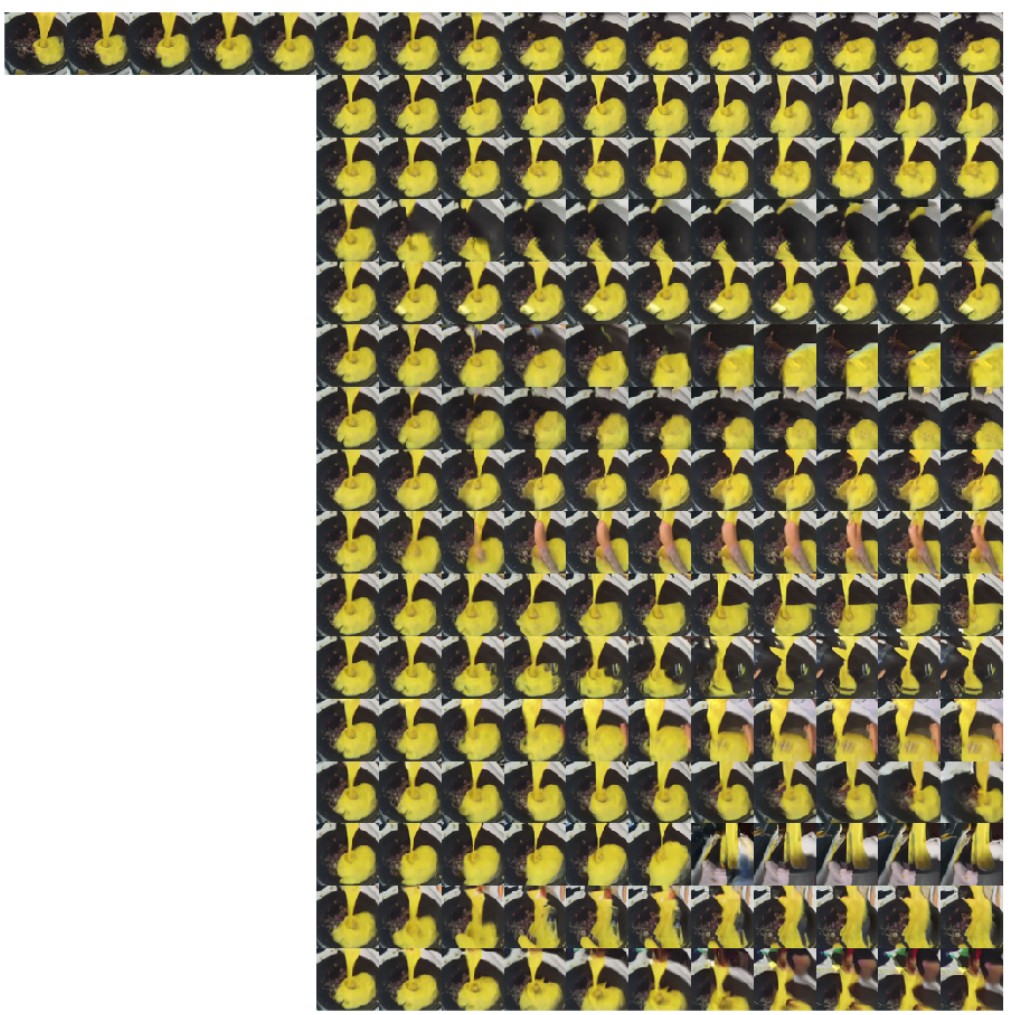

Figure 11: Samples of 11 future frames from the spatiotemporal subscaling model with 5 prime frames on 64x64 Kinetics.

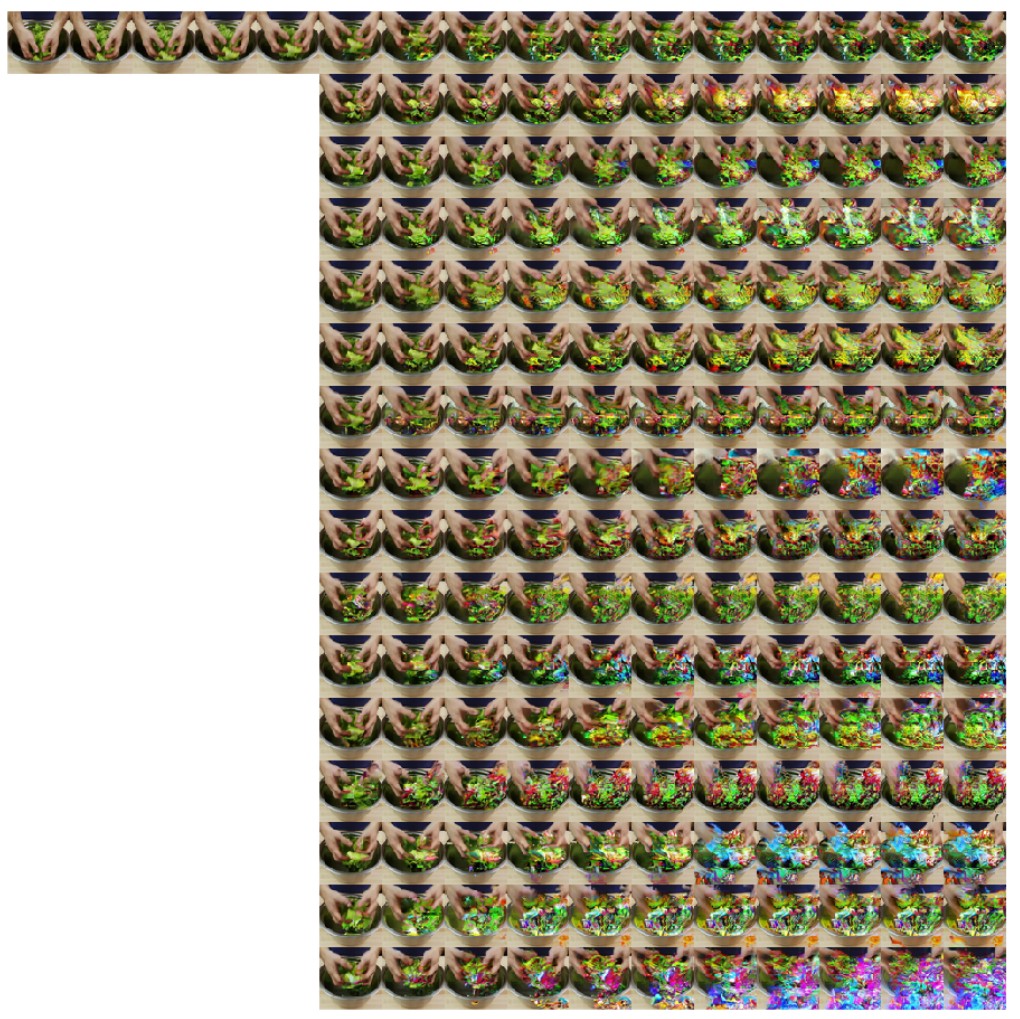

Figure 12: Samples of 11 future frames from the single frame model with 5 prime frames on 64x64 Kinetics exhibiting strange color artifacts.

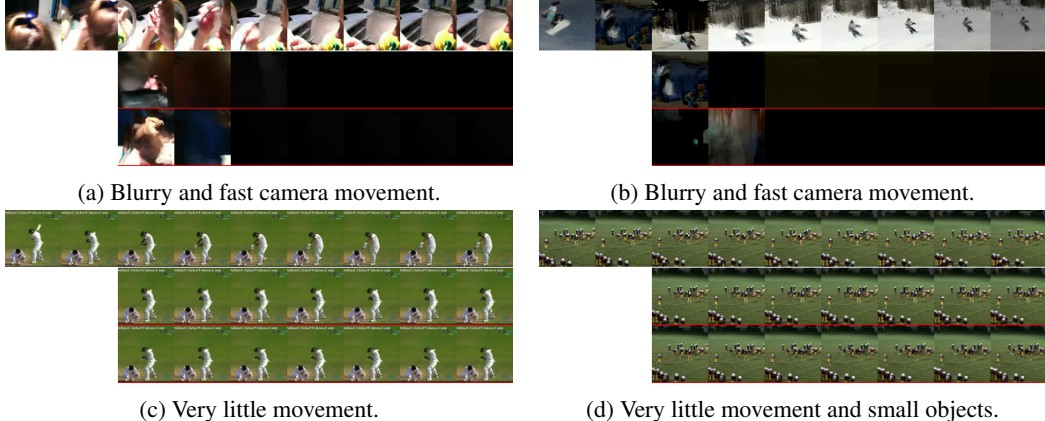

(a) Blurry and fast camera movement.

(b) Blurry and fast camera movement.

(c) Very little movement.

(d) Very little movement and small objects.

Figure 13: Ground-truth (top) and 2 samples of 30 future frames (showing every 4th frame) demonstrating that random Kinetics videos do not always lend themselves as good prefixes for generating continuations.

