# OpenReview forum: "Scaling Autoregressive Video Models"
_ICLR.cc/2020/Conference — Accept (Spotlight)_

### Official Review · AnonReviewer1 · 2019-10-20
**Official Blind Review #1**

**Rating:** 8

**Review:**


Summary
This papers presents a pixel-autoregressive model for video generation, in the spirit of VPN (Kalchbrenner’16). The proposed method uses video transformers and is made computationally efficient by extending block-local attention (Parmar’18, Chen’18) and sub-scaling (Menick’19) to 3D volumes. The block-local attention is separable, meaning that in theory it is possible to connect every two pixels through a sequence of block-local layers. However, for efficient parallelization implement via masking mechanism it is necessary to ignore certain connections, introducing independence assumptions.  The model is shown to substantially exceed state-of-the-art in terms of likelihood as well as quantitative and qualitative visual quality on several datasets, including the very challenging Kinetics-600. Interestingly, it is shown that the model with spatiotemporal subscaling is more robust to higher generation temperatures, which could imply robustness to accumulating errors.

Decision
The paper proposes a well-motivated method backed by solid state-of-the-art results. I recommend accept.

Pros
- The proposed method is relevant and well-motivated.
- The experimental results are strong.

Cons
- The paper novelty is somewhat limited as it is mostly a combination of previously existing techniques.
- The paper does not provide code which makes the results not easily reproducible. I think a minimal example of the code should be provided that is trainable at least on a simple dataset.

Questions
- No videos are provided. Please provide an (anonymous immutable) link to video results.
- Strong aliasing artifacts can be seen in the supplement on the Kinetics data, such as vegetables becoming increasingly “blocky” as well as general cube-like aliasing artifacts in Fig. 9. This indicates that the introduced independence assumptions are likely hurting the video quality. The paper discusses this in the appendix C, stating that there seems to be no remedy for the independence assumptions that does not increase the computational cost. However, this is exactly the problem that latent variable models such as variational inference or normalizing flows are designed to address. Would a certain combination of latent variable models with the proposed autoregressive approach alleviate these issues?

Minor comments
- Contrary to the summary in the related work section, Kumar’19 does not use variational inference and operates purely on the normalizing flows technique. Similarly, Mathieu’16 and Vondrick’16 do not use variational inference either instead relying on adversarial techniques. The paper correctly states that Lee’18, Castrejon’19 use variational inference.
- Figure 2 is never referred to in the text.


**Experience Assessment:**

I have published one or two papers in this area.

**Review Assessment: Checking Correctness Of Derivations And Theory:**

N/A

**Review Assessment: Checking Correctness Of Experiments:**

I carefully checked the experiments.

**Review Assessment: Thoroughness In Paper Reading:**

I read the paper thoroughly.

---

> ### Author Response · Authors · 2019-11-11
> **RE: Official Blind Review #1**
>
> We thank the reviewer for detailed review and comments. We will try to answer open questions as best as possible in the following.
>
> == Open Source Code ==
> We aim to open source code as soon as possible, hopefully in time for the publication.
>
> == Videos ==
> A link to videos is actually provided in the beginning of section 4.
>
> == Corrections ==
> We are sorry for the incorrect associations in our related work section and will update the paper as soon as possible. We will also make sure to mention Figure 2a in the main text.
>
> == Aliasing effects ==
> Indeed these effects are visible. However, the per-frame systematic independences are probably not the (sole) cause, as there should actually be very few. If it was, a slightly adapted model, using a masked CNN of slightly larger spatial receptive field should be able to prevent such unfortunate behaviour. However, we believe that this is due to the fact that indirect connections between pixels over multiple layers might sometimes be too weak to establish dependence. Interestingly, though, we did not observe any visual deterioration on BAIR robotic pushing, indicating that these effects might actually be due to under-parameterization. In fact, we believe that we might need to scale models even further to achieve strong performances on datasets of the complexity of Kinetics.
>
> ==  Combination with VAEs or normalizing flows ==
> Combining AR models with more complicated approaches such as VAEs or normalizing flows to alleviate the systematic independence problem would indeed be an interesting research direction. However, note that it is possible to design AR models (e.g., Image Transformer for images) that do not exhibit such independence assumptions. Unfortunately, we did not find a viable solution that scaled well to TPUs by the time of writing this paper.

---

> > ### Comment · AnonReviewer1 · 2019-11-11
> > **The response is satisfactory**
> >
> > Thanks for the response!
> >
> > - Thanks for going through the effort to open-source the code!
> >
> > - I indeed somehow missed the videos, thanks for pointing out the link!
> >
> > == Aliasing effects ==
> > I agree that the effects are visible to a much smaller extent on the BAIR dataset, which indeed very likely indicates underfitting. The explanation seems plausible. I want to note that I think the effects are still visible on BAIR, such as in Fig 8, rows 6,7 from the top, the shadow border becomes a straight line over time.
> >
> > ==  Combination with VAEs or normalizing flows ==
> > The response convinces me that the independence assumptions might not be the most pressing problem. The problem of learning spatially or temporally distant dependencies seems to be a much harder one, and current latent variable approaches also commonly suffer from this problem.

---

> ### Author Response · Authors · 2019-11-14
> **Corrections.**
>
> We updated our related work section slightly so that it becomes clearer that many of the mentioned works (after discussing VAE based approaches) are actually completely different directions and not additions upon VAEs.
>
> We added a reference for Figure 2 in section 4.2, paragraph "Qualitative Observations".

---

### Official Review · AnonReviewer3 · 2019-10-23
**Official Blind Review #3**

**Rating:** 8

**Review:**

This paper presents an approach for scalable autoregressive models for video synthesis. Key to the approach is a form of 3D (2 space and one time) self-attention that operates in a softly local manner (through a bias on the attention weights that makes them tend to prefer nearby connections), and also limits its field of view to a specific 3D sub-"block" of video at each layer for scalability. They also propose a clever ordering for autoregressive synthesis of the video subsampling spatially and temporally to generate multiple slices that are synthesized autoregressively one after the other. Each of these ideas is individually close to ideas proposed elsewhere before in other forms, as the authors themselves acknowledge [Vaswani et al 2017, Parmar et al 2018, Parikh et al 2016, Menick et al 2019], but this paper does the important engineering work of selecting and combining these ideas in this specific video synthesis problem setting.

Results on standard datasets for video generation match up to and/or surpass prior methods, in line with prior work on autoregressive image generation that has been shown to do very well on similar metrics (perplexity and FID). What is perhaps more interesting is that this paper presents initial promising results for open-world Youtube video settings (Kinetics dataset) that have not been evaluated systematically in any prior work in this area, to my knowledge.

The downsides of this paper are largely common to this method class (autoregressive generative models): training time (one of their models is "trained in parallel on 128 TPU v3 instances for 1M steps"), inference time (four short 64x64 video clips of 30 frames take 8 mins to generate on a Tesla V100), and model sizes (373M parameters for the Kinetics model). However, this does not take away from the contributions made here, that make it possible at all to train an autoregressive model of this size.

On the experiments, some questions, comments, and suggestions that the authors might consider addressing:
- How well do methods like SVG, SAVP, SV2P do on Kinetics, for comparison? It would be still more interesting if those models were scaled to have similar sizes to the large model in this work. While these methods have never been evaluated before on such unconstrained data, it is not clearly established that they do not work at all.
- To what extent does the blocking help, and when does it breaks down? e.g. how many layers/how large do blocks have to be for the idea of using different block sizes to suffice for smooth video synthesis? What happens when the blocking idea is not used at all?
- Other choices that aren't ablated in experiments: the choice of a local preference using the bias term in attention, the Transformer-style multi-attention heads. I do understand that these models are expensive to train and evaluate, but perhaps a smaller dataset might still suffice to demonstrate the value of these choices.
- Why is the proposed approach evaluated only on video prediction? Could it not be used for video generation without conditioning or with class conditioning?
- It is surprising to me that the perplexity of Kinetics models is lower than BAIR. Is there a reasonable explanation?

Writing and presentation are good for the most part, despite the main paper being dense with details and multiple fairly involved ideas. I particularly enjoyed parts of related work, the illustration of slicing in Fig 1, and the illustrative examples in Fig 3.

I would suggest however, that the paper might benefit from placing Sec 3.2 which describes the framework, before Sec 3.1. Fig 1 also belongs closer to Sec 3.2 anyway.

There are also terms/phrases I don't understand despite being reasonably familiar with the field like "positional embbeddings" (Sec 3.2). I also don't understand the need for "one-hot encoding of the discretized pixel intensities" (in that same paragraph). As a more minor comment, a footnote 1 before Eq 1 declares that capital letters denote matrices right before using capital letters to denote constants (T, H, W etc.).

**Experience Assessment:**

I have published one or two papers in this area.

**Review Assessment: Checking Correctness Of Derivations And Theory:**

N/A

**Review Assessment: Checking Correctness Of Experiments:**

I assessed the sensibility of the experiments.

**Review Assessment: Thoroughness In Paper Reading:**

I read the paper at least twice and used my best judgement in assessing the paper.

---

> ### Author Response · Authors · 2019-11-11
> **Re: Official Blind Review #3**
>
> Thank you for your detailed comments and recommendations for improvement in clarity! We will try to clarify the terms in the final version.
>
> == Model size, training- and generation time ==
> We would like to point out that training time is not a problem specific to autoregressive models - on the contrary since there are no latent variables to infer and no recurrence to unroll, fully attention-based autoregressive models are among the most efficient to train as gradient computation is completely parallel across the 3D volume during training.
>
> We agree that generation time is currently a considerable practical limitation of these methods. However, as mentioned in the paper, we believe that parallel generation methods (e.g., Stern et al., 2018) and low-latency hardware could bring down this substantially.
>
> Regarding model-size, note that VideoFlow (Kumar et al., 2019), for instance, has about the same number of parameters than our base models, yet our perplexity is much lower, our generated videos have much better fidelity and maintain long-range temporal dependencies (e.g., objects hidden by the robot arm for multiple frames) better.
>
> UPDATE: The authors of VideoFlow corrected their initial response to us regarding model size. Though still slightly higher, it turns out that their models' number of parameters are in the same ball park as our base models. Hence, we corrected our statements above.
>
> == Comparison on Kinetics ==
> Our aim with this work is to push the limits of autoregressive models and demonstrate their effectiveness as baselines for competitive video prediction, as illustrated by the experiments on BAIR, while also providing momentum towards exploring much more challenging tasks. We would definitely be interested in seeing how other methods would perform on the Kinetics dataset and hope that the community will take on this challenge.
>
> == Block-local attention ==
> Block-local attention is necessary to limit memory requirements as attention is quadratic in the number of pixels, which grows prohibitive for 3D volumes. Block-local attention brings this down to linear complexity, similarly to the concurrently proposed flattened sparse attention of Child et al. (2019), while maintaining the explicit 3D structure of videos and not requiring any custom kernels.
>
> We did experiment with varying block sizes in the time- and space dimensions across different layers and found the model robust to this choice. It seems that what matters is that there is a sufficient connectivity between pixels across the video volume, rather than the exact choice of per-layer connectivity.
>
> == Ablation and relative-position prior ==
> To clarify, the relative position attention-bias term does not enforce a local preference - it is a learned parameter which simply gives the model capacity to take relative position information into account. Note that we do ablate the number of attention heads, number of layers and the hidden size in Table 3 of the appendix, where we find the hidden size to be the most effective way to improve perplexity.
>
> == Why focus on video prediction ==
> This is an interesting question. We have focused on video generation conditioning on an initial frame to stay comparable to existing work. Completely free video generation is much more difficult to evaluate, beyond visual inspection and perplexity. By conditioning on an initial frame, measures such as FVD are much more informative. We also believe that predicting future frames is a more practically interesting task. However, we believe that autoregressive models could be competitive for unconditional generation as well, following the results on unconditional image generation by Menick & Kalchbrenner (2019).
>
> == Lower perplexity on Kinetics ==
> This is indeed an intriguing finding. There are many potential reasons. However, we think this might be due to the lower frame-rate in the BAIR robotic pushing dataset (10 frames per second) compared to Kinetics (25), resulting in faster movement between frames.

---

### Official Review · AnonReviewer2 · 2019-10-24
**Official Blind Review #2**

**Rating:** 6

**Review:**

This work proposes an autoregressive video generation model, which is based on a newly proposed three-dimensional self-attention mechanism. It generalizes the Transformer architecture of Vaswani et al. (2017) to spatiotemporal data (video). The original Transform implies self-attention among different words in a sentence. Considering the larger scale of video, this work proposes to divide it into small blocks, and apply the self-attention (part of block-local self-attention modular) on each block. At the same time, it addresses the information exchange between blocks problem, by spatiotemporal sub-scaling (described in section 3.2). The proposed method achieves competitive results across multiple metrics on popular benchmark datasets (BAIR Robot Pushing and KINETICS), for which they produce continuations of high fidelity.

Some questions:
-      The proposed model is claimed to work on competitive results across multiple metrics on popular benchmark datasets. However, it only compares with stat-of-the-art models on BAIR Robot Pushing dataset (for the other dataset, the author only compares with the variations of the proposed model). Further, the author only reports the result of Bits/dim and FVD, instead of PSNR and SSIM, which are reported in the original papers. Any justification for this? Though FVD has its own advantages, showing PSNR and/or SSIM at the same time would help us get better sense of the performance.
-      In table 1 (left) in section 4.2, the author mentions that results from all the stat-of-the-art models are not strictly comparable, since prior works use longer priming sequences of two or three frames, whereas the proposed models only observe a single prime frame. I am confused of why the proposed model can only see a single frame.
-      The proposed block-local self-attention modular works on divideding video into small blocks, which seems to be a matrix of 3 or 4 dimensions (t,h,w,c). However, in the experiment, the input of the model for the BAIR Robot Pushing dataset is the first frame. How can this frame be feed into the block-local self-attention modular?
-     In section 3.3, it splits the 3x8-bit RGB channel into 6x4-bit channels. It would be better if the author can show an example and clarify the advantages.
-     In section 3.3, U_k, N_v and P seems not defined in the context.
-    Videos are divided into small blocks and feed into the block-local self-attention modular separately. Then, I’m confused on how to aggregate these different blocks together to predict future frames.

I would like to raise up my score if the author can address my questions.


**Experience Assessment:**

I have read many papers in this area.

**Review Assessment: Checking Correctness Of Derivations And Theory:**

I assessed the sensibility of the derivations and theory.

**Review Assessment: Checking Correctness Of Experiments:**

I assessed the sensibility of the experiments.

**Review Assessment: Thoroughness In Paper Reading:**

I read the paper at least twice and used my best judgement in assessing the paper.

---

> ### Author Response · Authors · 2019-11-11
> **RE: Official Blind Review #2**
>
> We thank the reviewer for their thorough review and try to address the questions and concerns in the following.
>
> == Comparison to prior work on Kinetics ==
> We would definitely be interested in seeing how these methods would perform on this dataset. However, setting up these methods for large-scale training and hyper-parameters tuning to allow for fair comparison would require a substantial engineering effort and time to experiment. Given the effort required to run all of the experiments in the paper, we determined this to be out scope for the current work. Our aim with this work is to push the limits of autoregressive models and demonstrate their effectiveness as baselines for competitive video prediction, as illustrated by the experiments on BAIR, while also providing momentum towards exploring much more challenging tasks.
>
> == PSNR and SSIM ==
> We justify not including PSNR and SSIM in the section 4.1 (Extrinsic Evaluation). To summarize, prior work has found that those metrics do not correlate well with perceptual quality (see for instance Lee et al., 2018, Unterthiner et al., 2019). In particular, VAEs perform very well on SSIM and PSNR on BAIR, despite the actual videos being quite blurry. In any case, we found that in terms of these metrics our models actually performed better on BAIR than SAVP, but still worse than VAE-based models. We discussed including those results but refrained from it in the interest of brevity and because they can be misleading, as demonstrated clearly by Lee et al., 2018.
>
> == Only single frame priming ==
> There have been discrepancies between evaluation protocols in prior work (e.g.,  between SAVP and VideoFlow). We settled for priming on a single frame because we found that our model was able to handle this (harder) setting quite well. Conditioning on 2 frames or more improves our scores slightly, but due to the inherent stochasticity of the robot arm, the difference is negligible.
>
> == How can the conditioning frame be fed into the block-local self-attention module? ==
> We actually do not really condition our model on initial frames using another prefix encoder. Instead we only have a single decoder for which we fix the given prefix pixels to the ground truth. This is referred to as "priming" in the paper. This means that the hidden representation of the initial priming frames are computed using the ground truth instead of generated pixels.
>
> == Channel Splitting ==
> The results of Menick & Kalchbrenner (2019)  for image generation suggested that encoding and decoding the first four bits for each channel prior to the four fine bits is important. The argument is that the fine bits are quite easy to predict conditioned on the coarse bits. However, in our experiments we found only slight improvements of this splitting. On the other hand, splitting channels do improve the memory footprint in the slice encoder (thanks to a smaller one-hot encoding), so we kept with this setup for all experiments.
>
> == How to aggregate different video blocks ==
> Videos are divided into blocks before each layer and subsequently reunited after each layer. Crucially the shapes of the blocks differ at every layer to allow for efficiently connecting pixels (indirectly) over large distances. For instance, consider a pixel at position [n,m] in a 2D image which we would like to connect to a pixel at position [1,1]. We can either do this directly if the block is large enough (at least of size [n,m]). However, this becomes intractable for large values of n,m due to the quadratic nature of attention. An alternative is to run one layer with self-attention in blocks of [1,m] --- which would connect pixel [1,m] to pixel [1,1] (among others of course) --- followed by self-attention in blocks of size [n,1] --- connecting pixel [n,m] with [1,m] which itself is already connected to [1,1]. This way, information can flow from pixel [1,1] to pixel [n,m] through pixel [1,m].

---

> ### Author Response · Authors · 2019-11-14
> **RE: Official Blind Review #2  Part 2**
>
> Additional clarification concerning the definition of U_k, N_v and P in section 3.3.
>
> We define the dimensionality of U_k and P in Eq. 8. These are trainable parameters.
>
> N_v is actually defined in the text as "N_v=16". It is 16 because we predict 6 channels ("N_c=6") of 4bit per channel. 4bit translates to 2^4=16 potential values.

---

### Decision · Program_Chairs · 2019-12-19

**Decision:**

Accept (Spotlight)

**Comment:**

This paper presents an approach for scalable autoregressive video generation based on a three-dimensional self-attention mechanism. As rightly pointed out by R3, the proposed approach ’is individually close to ideas proposed elsewhere before in other forms ... but this paper does the important engineering work of selecting and combining these ideas in this specific video synthesis problem setting.’
The proposed method is relevant and well-motivated, and the experimental results are strong. All reviewers agree that experiments on the Kinetics dataset are particularly appealing. In the initial evaluation, the reviewers have raised several concerns such as performance metrics, ablation study, training time comparison, empirical evaluation of the baseline methods on Kinetics, that were addressed by the authors in the rebuttal.
In conclusion, all three reviewers were convinced by the author’s rebuttal, and AC recommends acceptance of this paper – congratulations to the authors!